# Immunosuppressive low-density neutrophils in the blood of cancer patients display a mature phenotype

Christophe Vanhaver[1,*], Frank Aboubakar Nana[1,2,3,*] , Nicolas Delhez[1] , Mathieu Luyckx[1,4,5], Thibault Hirsch[1] , Alexandre Bayard[1], Camille Houbion[1], Nicolas Dauguet[1], Alice Brochier[6], Pierre van der Bruggen[1] , Annika M Bruger[1]

The presence of human neutrophils in the tumor microenvironment is strongly correlated to poor overall survival. Most previous studies have focused on the immunosuppressive capacities of low-density neutrophils (LDN), also referred to as granulocytic myeloid-derived suppressor cells, which are elevated in number in the blood of many cancer patients. We observed two types of LDN in the blood of lung cancer and ovarian carcinoma patients: CD45$^{high}$ LDN, which suppressed T-cell proliferation and displayed mature morphology, and CD45$^{low}$ LDN, which were immature and non-suppressive. We simultaneously evaluated the classical normal-density neutrophils (NDN) and, when available, tumor-associated neutrophils. We observed that NDN from cancer patients suppressed T-cell proliferation, and NDN from healthy donors did not, despite few transcriptomic differences. Hence, the immunosuppression mediated by neutrophils in the blood of cancer patients is not dependent on the cells' density but rather on their maturity.

## Introduction

Neutrophils, the most abundant leukocytes in the blood, are the first responders to sterile injury and microbial infection (Borregaard, 2010). The presence of neutrophils within the tumor microenvironment is more strongly correlated with unfavorable outcomes for patients than the presence of any other leukocyte (Jensen et al, 2012; Gentles et al, 2015; Ye et al, 2019). However, both antitumor and protumor functions of tumor-associated neutrophils (TAN) have been described (Shaul & Fridlender, 2021; Gungabeesoon et al, 2023). In addition, elevated levels of low-density neutrophils (LDN) in the blood indicate poorer treatment outcomes and survival prognoses for cancer patients (Sade-Feldman et al, 2016; Lang et al, 2018; Wang et al, 2018). Focusing on protumor neutrophils in cancer patients

could provide new strategies to enhance current immunotherapies, help predict the therapies' efficacy, or help clinicians individualize treatment plans (Nagaraj et al, 2010; Moses & Brandau, 2016; Shaul et al, 2020; Gondois-Rey et al, 2021).

LDN are more frequent in the blood of cancer patients than in healthy donors (Cassetta et al, 2020). However, isolation (e.g., flow cytometry versus magnetic bead selection) and phenotyping strategies differ between studies (Cassetta et al, 2019). A multicenter study by the COST Action Mye-EUNITER applied the same staining and gating strategy to blood samples from patients with various cancer and autoimmune conditions, and found that LDN are more present in the blood of patients with some cancer types, including melanoma, compared with others, for example, ovarian cancers (Cassetta et al, 2020). Studying neutrophils remains demanding because these cells have an average half-life of 18 h in humans and are cryosensitive (Tak et al, 2013; Trellakis et al, 2013; Lahoz-Beneytez et al, 2016).

The paradigm that neutrophils are an entirely homogenous population with unique and specific functions has been overturned (reviewed in Scapini et al [2016], Shaul and Fridlender [2018], and Zilionis et al [2019]). Instead, neutrophils respond to environmental cues and adopt tissue-specific features (Mackey et al, 2019; Ballesteros et al, 2020; Montaldo et al, 2022). Neutrophil subsets seem to fall along a single differentiation pathway, meaning that subsets share many characteristics and may only be distinguished by properties such as density, maturity, and functions (Scapini et al, 2016; Blanter et al, 2021).

Here, we compared the presence, transcriptome, and function of blood neutrophils and TAN. In blood, we compared classic neutrophils (the majority of circulating neutrophils, hereafter referred to as normal-density neutrophils [NDN]) and LDN. We provide additional insights into the ability of distinct neutrophil populations to suppress T-cell proliferation in vitro, which serves as a criterion for classifying a neutrophil as a granulocytic myeloid-derived suppressor cell. Detailed single-cell transcriptomic analyses, such as those conducted by Pittet and Trajanovski (Zilionis et al, 2019; Salcher et al, 2022), have uncovered the diversity of neutrophils within tumors of cancer

---

[1]Institut de Duve, Université Catholique de Louvain, Brussels, Belgium   [2]Service de Pneumologie, Cliniques Universitaires Saint-Luc, Brussels, Belgium   [3]Institut de Recherche Expérimentale et Clinique (IREC)/Pôle de Pneumologie, Université Catholique de Louvain, Brussels, Belgium   [4]Service de Gynécologie et Andrologie, Cliniques Universitaires Saint-Luc, Brussels, Belgium   [5]Centre de Chirurgie Oncologique, Institut Roi Albert II, Cliniques Universitaires Saint-Luc, Brussels, Belgium   [6]Hematology Department of Laboratory Medicine, Cliniques Universitaires Saint-Luc, Brussels, Belgium

Correspondence: pierre.vanderbruggen@uclouvain.be
*Christophe Vanhaver and Frank Aboubakar Nana contributed equally to this work

patients. We nevertheless deliberately opted for bulk analyses to assess both the suppressive capabilities and transcriptome of the same neutrophil sample.

## Results

### Both NDN and LDN of NSCLC patients suppress T-cell proliferation

We sourced blood and tumor from either NSCLC or ovarian cancer patients, and blood from non-cancerous donors as healthy controls. No patient had received chemotherapy during the year before the collection of blood or tumor samples, nor had they taken corticosteroids, except for patient A017 in Fig 2. NDN and mononuclear cells were isolated from blood for flow cytometric analysis or sorting following the published strategy (Bronte et al, 2016; Bruderek et al, 2021) (Fig 1A and B). We rarely obtained enough TAN for further experimentation.

LDN-mediated suppression of T-cell proliferation in vitro is most commonly investigated by performing co-incubation assays of T cells with LDN of cancer patients (Haile et al, 2012; Bruger et al, 2020). We also tested NDN from the same patients and from healthy donors as a functional control. The LDN and NDN were co-incubated with allogeneic or autologous T cells for 4 d. T-cell proliferation was assessed by flow cytometry using the dilution of a tracker dye (Fig 1C).

We observed that LDN isolated from NSCLC cancer patients suppressed both allogeneic (Fig 1D and E) and autologous (Fig S1) T-cell proliferation. However, immunosuppression by LDN varied across patient samples. NDN from cancer patients suppressed T-cell proliferation (Fig 1D and E), whereas NDN from healthy donors did not (Fig S1B). A few NSCLC tumors were available. They were enzymatically and mechanically digested, and TAN were isolated from single-cell suspensions using the same method as for blood LDN. Two of the six samples also exhibited suppression of T-cell proliferation (Fig 1D and E).

Compared with blood from NSCLC patients, LDN were rare in the blood of ovarian cancer patients (often 0.01% of CD45-positive cells) and healthy donors (Fig S2A–D). These frequencies are consistent with the frequencies reported (Cassetta et al, 2020). LDN yields from ovarian cancer blood samples were often too low (1,000–10,000 cells) for further functional experimentation. NDN and LDN isolated from ovarian cancer patients also reduced allogeneic T-cell proliferation, but statistical significance was not reached because of the low sample size (Fig S1C).

Altogether, our data indicate that immunosuppression is a common feature of neutrophils in the blood of cancer patients: when LDN were suppressive, the corresponding NDN were also suppressive. In fact, the suppressive activity of NDN was systematically more pronounced.

### Two LDN subsets exist in the blood of cancer patients, and they can be distinguished from each other based on the expression of CD45, maturity and immunosuppressive function

We observed that some LDN samples suppressed T-cell proliferation to a high degree, whereas some LDN samples had no effect on T-cell proliferation at all. This was particularly evident in the blood of NSCLC patient A017 whose blood was sampled three times within 2 mo. At the first sampling, the patient's LDN suppressed T-cell proliferation very strongly. The second time, when the patient was administered corticosteroid and methylprednisolone, LDN exhibited no suppressive capacity (Fig 2A). Upon closer inspection of the samples analyzed by flow cytometry, we observed the presence of two LDN subtypes with similar granularity (SSC-A) distinguishable by the expression of CD45, and whose frequency changed between the two visits (Fig 2B). We used the blood collected during the third visit of the patient to isolate CD45$^{high}$ LDN, CD45$^{low}$ LDN, and all LDN and NDN. CD45$^{low}$ LDN had no suppressive effect on T-cell proliferation, whereas the CD45$^{high}$ LDN and the NDN were both highly suppressive (Fig 2C). To further determine the differences between the neutrophil subsets, we performed cytospin analyses on blood and tumor neutrophil subsets (Fig 2D). We observed that CD45$^{low}$ LDN are comprised of immature myelocytes, metamyelocytes, and some band cells. CD45$^{high}$ LDN, NDN, and TAN samples contain nearly exclusively mature neutrophils (Fig 2D).

When we re-analyzed the blood and tumor samples' phenotyping analyses that were previously performed, we found that the frequency of CD45$^{high}$ cells in the LDN population was highly variable, whereas about 80% TAN in NSCLC patients were CD45$^{high}$ (Fig 3A). When the T-cell proliferation assays previously performed with a mixed population of LDN (CD45$^{high}$ and CD45$^{low}$ cells) were re-analyzed, we noticed that the frequency of CD45$^{high}$ cells in a sample was positively correlated with the suppression capacity of that sample. This was true for both proliferation assays performed with allogeneic and autologous T cells (Fig 3B).

We subsequently sorted by flow cytometry CD45$^{high}$ LDN and CD45$^{low}$ LDN from the blood of NSCLC patients, and assessed their suppressive activity in proliferation assays with allogeneic T cells. We found that most CD45$^{high}$ LDN samples contained suppressive cells, whereas CD45$^{low}$ LDN samples slightly increased T-cell proliferation (Fig 3C and D). The same was observed with autologous T cells, except for one CD45$^{low}$ LDN sample that reduced T-cell proliferation (Fig S1D). Taken together, this indicates that mature LDN isolated from cancer patients suppress T-cell proliferation. Combined with the former observation that NDN from cancer patients are also mature neutrophils with suppressive properties, our data indicate that immunosuppressive neutrophils in cancer patients have a mature phenotype.

### CD45$^{low}$ LDN express higher levels of genes encoding ORL1, ARG1, and CYBB than suppressive CD45$^{high}$ LDN, but show similar levels of the corresponding proteins

We distinguished immature CD45$^{low}$ LDN and mature CD45$^{high}$ LDN from each other based on their expression of CD45 and CD16 (which is also highly expressed on mature neutrophils, including NDN) (Fujimoto et al, 2000; Elghetany, 2002; Lang et al, 2018; Gondois-Rey et al, 2021) (Fig 4A). From each NSCLC patient, we isolated at least 10,000 cells of the two cell types from the same blood sample and performed RNA sequencing. The heatmap of differentially expressed transcription factors revealed distinct transcriptional programs between immature CD45$^{low}$ LDN and

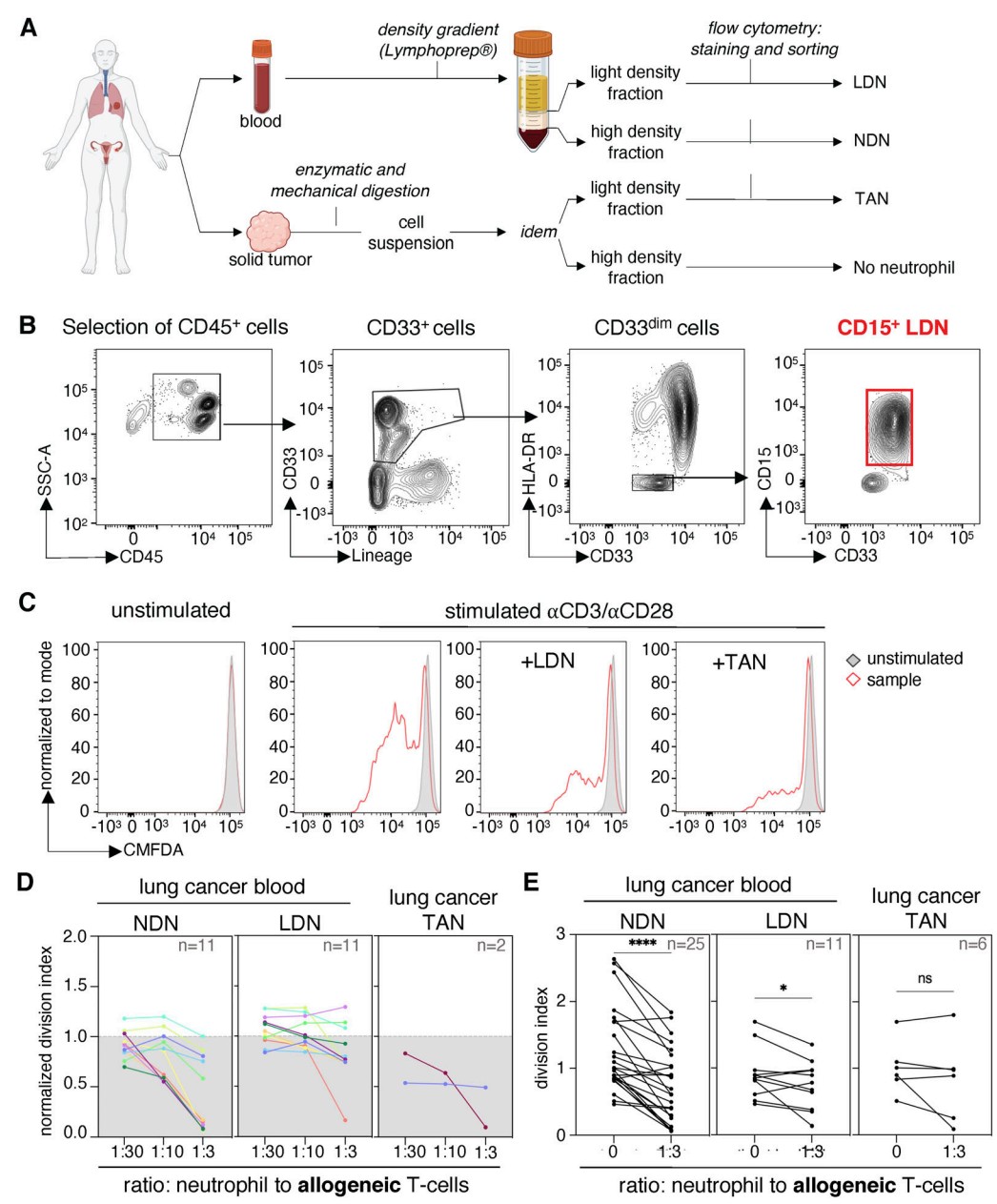

**Figure 1. Normal-density neutrophils (NDN) of NSCLC patients suppress T-cell proliferation more consistently than low-density neutrophils (LDN) and tumor-associated neutrophils (TAN).**
**(A)** Scheme of sample processing and preparation. **(B)** Gating strategy for the isolation of LDN. **(C)** Blood LDN and TAN isolated from patient A069 suppress proliferation of allogeneic T cells. 10,000 neutrophils (LDN or TAN) were incubated with 30,000 chloromethylfluorescein diacetate-labeled allogeneic T cells, with a ratio of 1 neutrophil to 3 T cells. The T cells were stimulated with coated anti-CD3 and soluble anti-CD28 antibodies for 4 d. **(D)** Suppression of T-cell proliferation by NDN, LDN, and TAN from NSCLC patients. The different patients are indicated by colors. T-cell proliferation was calculated using the division index and normalized against stimulated control samples without myeloid cells (control = 1). In (D), data from patients from whom we obtained both NDN and LDN subpopulations. **(E)** Statistics for all proliferation assays performed using the division index without normalization. The statistical analysis pertains to a ratio of 1 neutrophil to 3 T cells compared with T cells stimulated without myeloid cells. Statistical analyses: Wilcoxon's matched-pairs signed-rank test. *P*-values: ns (non-significant), ≥0.05; *, <0.05; and ****, <0.0001.

mature CD45$^{high}$ LDN (Fig S3). We confirmed that the expression of the CD45 and CD16-encoding genes matches the expression of the flow cytometry data acquired during the isolation of the cells (Fig 4B and C).

PD-L1, arginase-1, NOX2, which generates reactive oxygen species (ROS), and LOX-1 are frequently associated with

increased suppressive activity of LDN, either as a marker or as proteins mediating suppression itself (Zhou et al, 2018; Vanhaver et al, 2021). Suppressive CD45$^{high}$ LDN expressed higher levels of *CD274* (encoding PD-L1) than immature CD45$^{low}$ LDN (Fig 4D). The frequency of PD-L1–positive LDN, nevertheless, was close to zero, both in CD45$^{high}$ and in CD45$^{low}$ LDN (Fig

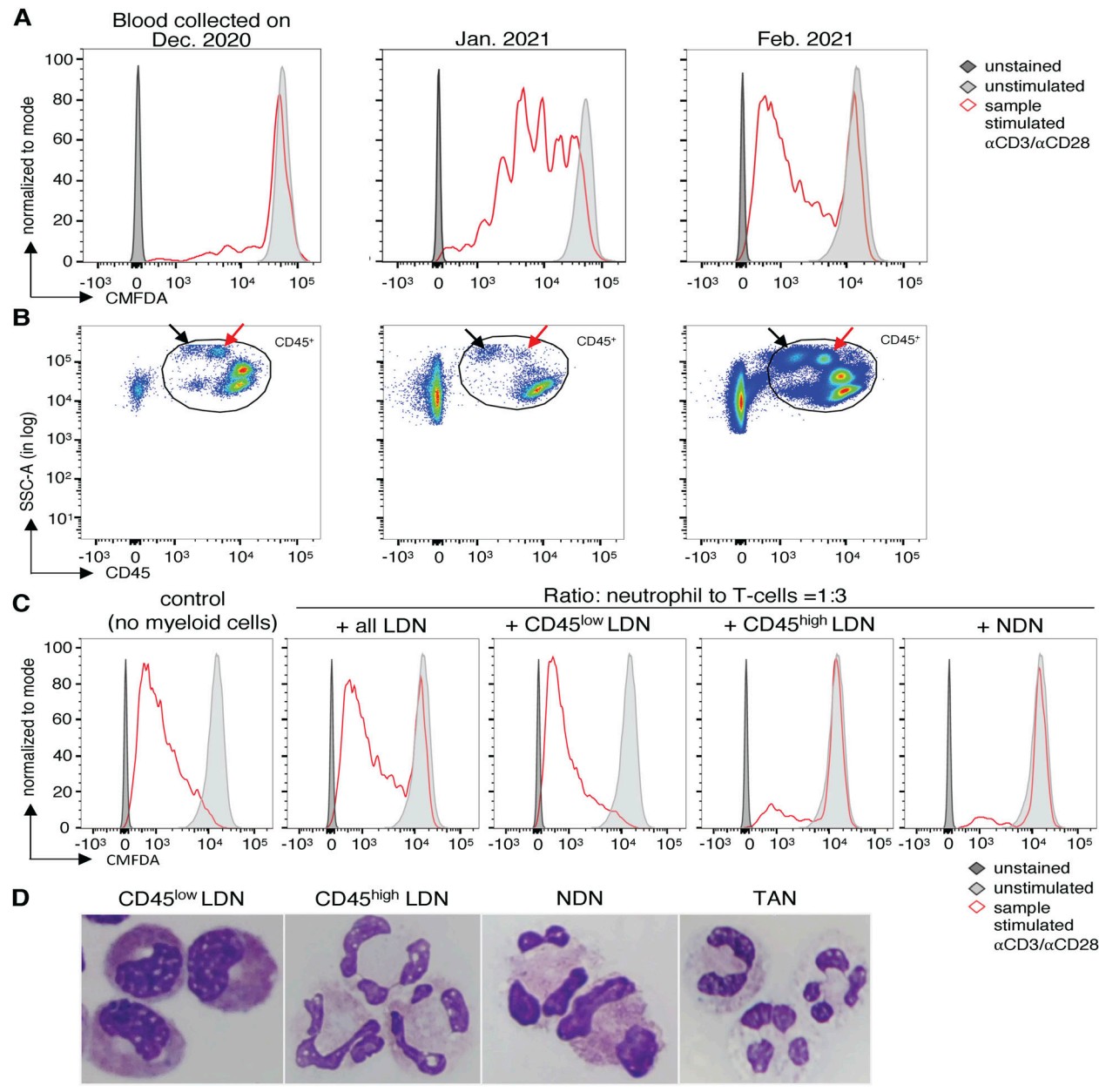

**Figure 2. Low-density neutrophils (LDN) of cancer patients consist of two subpopulations distinguishable by maturity and the expression of CD45.**
**(A)** Effect of all LDN of NSCLC patient A017 on the proliferation of allogeneic T cells. 30,000 chloromethylfluorescein diacetate-labeled allogeneic T cells were incubated with 10,000 LDN. The T cells were stimulated with coated anti-CD3 and soluble anti-CD28 antibodies for 4 d. **(B)** Flow cytometry plots on PBMC of NSCLC patient A017 on three dates. He received methylprednisolone from December 19, 2020, to February 07, 2021. Arrows indicate the two LDN subtypes distinguishable by the level of CD45 expression. **(C)** Detailed analysis of the effect of LDN subtypes and normal-density neutrophils isolated from NSCLC patient A017 on allogeneic T-cell proliferation in February 2021 (third sampling). The proliferation assay was conducted as described before. **(D)** Cytospin analysis of different populations of neutrophils of another patient. The different neutrophils were isolated on the same day from the same patient. Cells were stained with the May–Grünwald–Giemsa stain and analyzed with a 100x magnification using cytospin.

5B). PD-L1 was also absent from the surface of NDN (Fig S4A and B).

On the contrary, immature CD45$^{low}$ LDN—which are devoid of suppressive activity—expressed higher levels of *ORL1* (encoding LOX-1), *ARG1* (encoding arginase-1), and *CYBB* (encoding NOX2) than suppressive mature CD45$^{high}$ LDN (Fig 4D). Except for LOX-1, protein

levels appear to contradict the transcriptomic data: intracellular staining for arginase-1 and ROS detection by a fluorescent probe showed similar levels both in CD45$^{high}$ and in CD45$^{low}$ LDN (Fig 5B and C). LOX-1 levels were slightly higher on the surface of non-suppressive CD45$^{low}$ LDN (Fig 5D and E) compared with the suppressive CD45$^{high}$ LDN, but nearly absent from the surface of NDN.

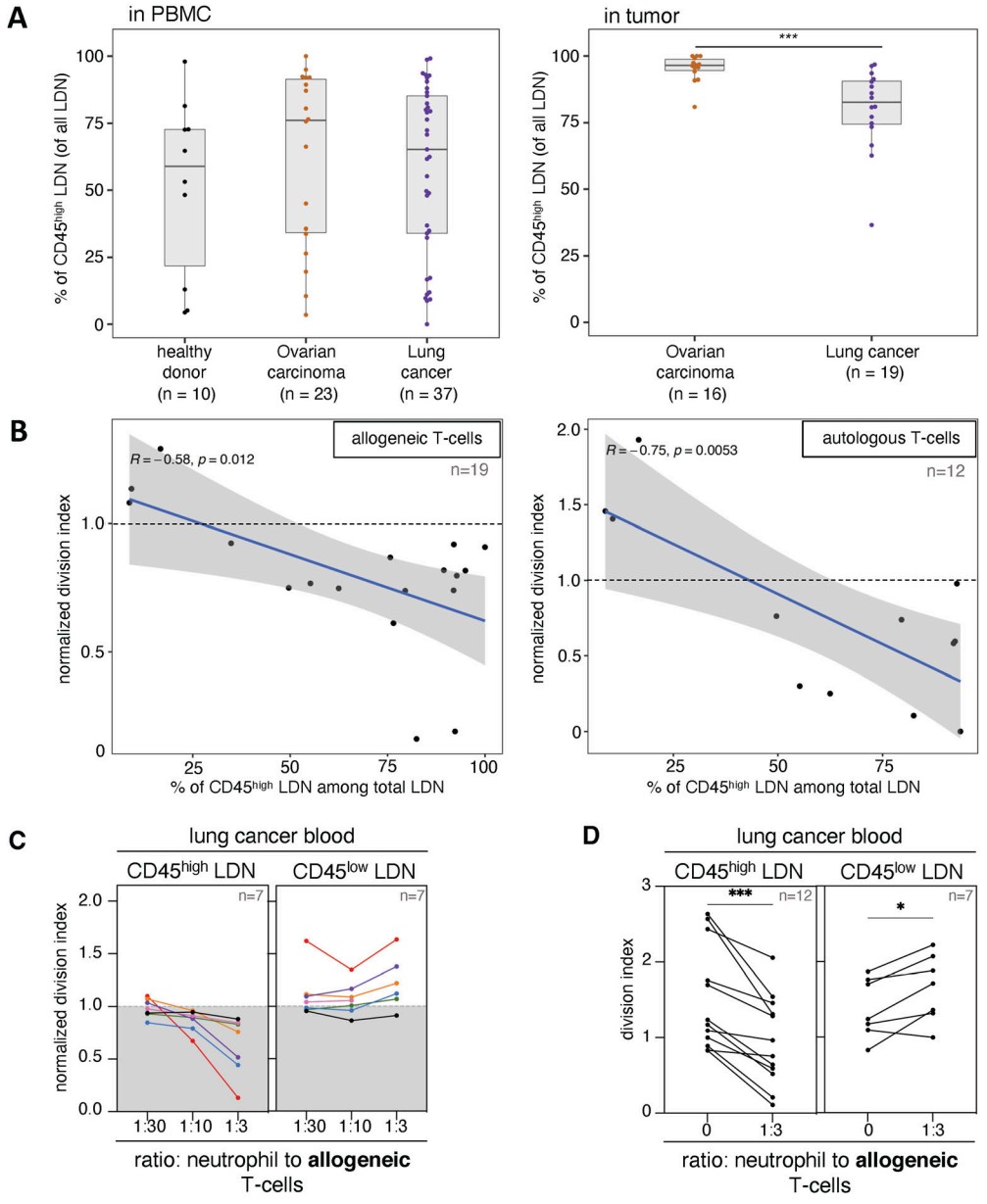

**Figure 3. Unlike CD45<sup>low</sup> low-density neutrophils (LDN), CD45<sup>high</sup> LDN of NSCLC patients suppress T-cell proliferation.**
**(A)** Frequency of CD45<sup>high</sup> LDN among all LDN in NSCLC blood and tumor samples. Boxes show the median, and whiskers, the 1.5× interquartile range. **(B)** Correlation of the frequency of CD45<sup>high</sup> LDN among all blood LDN with suppression of T-cell proliferation in autologous and allogeneic co-culture assays (1 LDN: 3 T cells). **(C)** Effect of CD45<sup>high</sup> or CD45<sup>low</sup> LDN from NSCLC blood on the proliferation of allogeneic T cells. The different patients are indicated by colors. Proliferation was calculated using the division index and normalized against stimulated control samples without myeloid cells (control = 1). In (C), data from patients from whom we obtained both neutrophil subpopulations. **(D)** Statistics for all proliferation assays performed using the division index without normalization. The statistical analysis pertains to a ratio of 1 neutrophil to 3 T cells compared with T cells stimulated without myeloid cells. Statistical analyses: Wilcoxon–Mann–Whitney's test (A) and Wilcoxon's matched-pairs signed-rank test (D). *P*-values: *, <0.05; and ***, <0.001.

As for the NDN, arginase-1 levels were equivalent to those of LDN. However, more ROS is detected in the NDN (Fig S4A and B).

### The transcriptomes of blood CD45<sup>high</sup> LDN and NDN exhibit high similarity, with slight differences from TAN

We next compared the transcriptomes of CD45<sup>high</sup> and CD45<sup>low</sup> LDN with those of NDN and TAN. In the resulting principal component analysis (PCA), immature CD45<sup>low</sup> LDN cluster apart from the other populations with the greatest distance, meaning that their transcriptional programs differ the most from the other neutrophil populations (Fig 6). This is further underlined in volcano plots (Fig S5A–D).

Before identifying the CD45<sup>high</sup> and CD45<sup>low</sup> subpopulations, we had compared the transcriptomes of bulk LDN and NDN from patients with lung or ovarian cancer. In the PCA plots (Fig S6A), the bulk LDN were at times closely clustered with the NDN, whereas at other times, they were not. We hypothesize that this variability might be influenced by the proportion of CD45<sup>high</sup> cells within the bulk LDN population. It is worth noting that the samples formed clusters irrespective of the cancer type (Fig S6B).

Genes associated with neutrophil granules, including *MPO* (myeloperoxidase), *MMP8* (matrix metalloproteinase 8), *CAMP* (cathelicidin antimicrobial peptide), *LTF* (lactotransferrin), *ARG1* (arginase-1), and *PRTN3* (proteinase 3), are up-regulated in immature CD45<sup>low</sup> LDN in comparison with mature CD45<sup>high</sup> LDN and NDN (Fig 7A and B). This is consistent with the immature state of CD45<sup>low</sup> LDN because primary

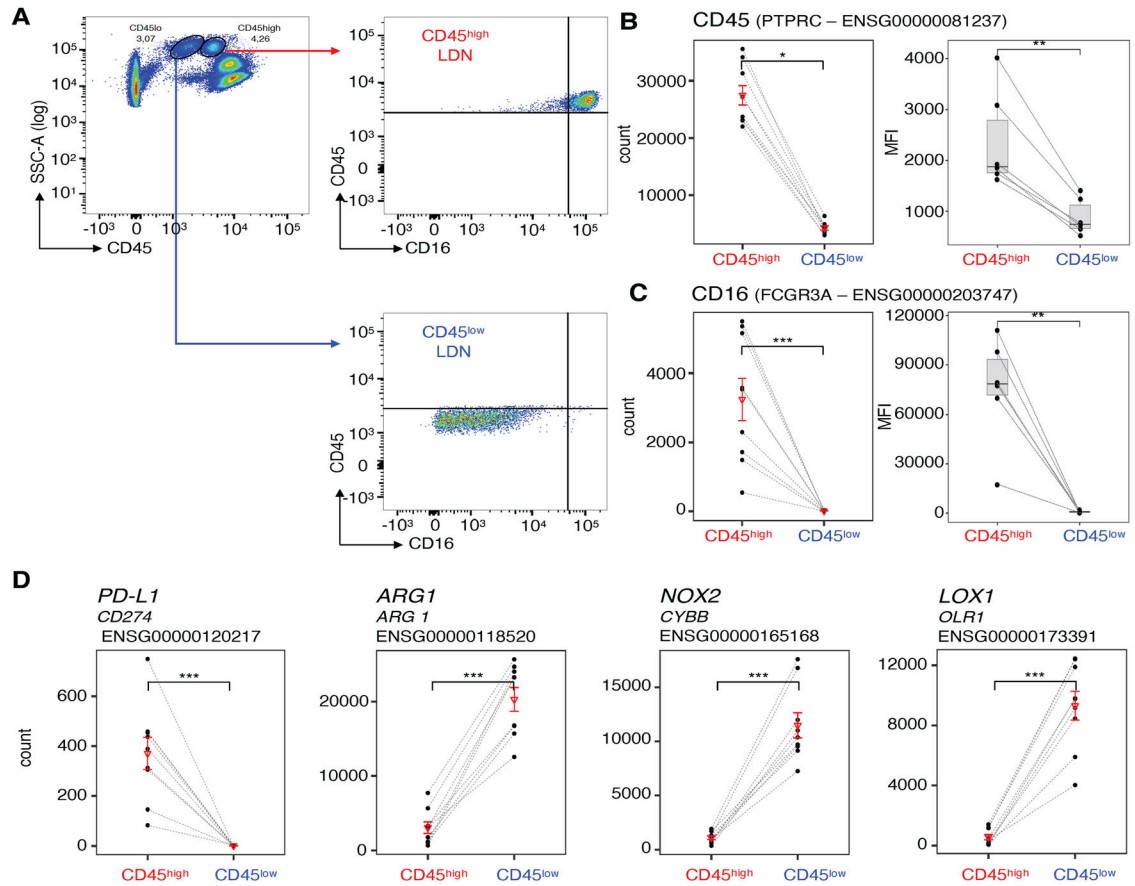

**Figure 4. Matched CD45^high and CD45^low low-density neutrophils (LDN) from NSCLC patients differentially express genes involved in neutrophil function and maturity.**

**(A)** Example of flow cytometric analysis of the differential expression of CD45 and CD16 by CD45^high and CD45^low LDN from the blood of one representative NSCLC patient. **(B, C)** Comparison between patient-matched CD45^high and CD45^low LDN from patients of read counts from the transcriptomic data (n = 9) and the mean fluorescence intensity (MFI) (n = 6) recorded by flow cytometry for (B) CD45 and (C) CD16. Boxes show the median, and whiskers, the 1.5× interquartile range. **(D)** Comparison of read counts from the transcriptomic data for genes associated with LDN-mediated immunosuppression (*CD274*, *NOX2*, *CYBB*) and *LOX-1* between patient-matched CD45^high and CD45^low LDN from the blood of NSCLC patients (n = 9) (DESeq2 test). Adjusted *P*-values for RNA-sequencing data. *P*-adj: *PTPRC* (count) = 0.028; *CD16* (count) = $1.93 \times 10^{-60}$; CD16 (MFI) = 0.031; CD45 (MFI) = 0.031; *OLR1* = $1.95 \times 10^{-6}$; *ARG1* = $2.644 \times 10^{-3}$; *CYBB* = $5.44 \times 10^{-7}$; *CD274* = $1.51 \times 10^{-22}$. Statistical analyses: Wilcoxon's matched-pairs signed-rank test.
*P*-values: *, <0.05; **, <0.01; and ***, <0.001. In graphs (B, C, D), the triangle corresponds to the mean and the line to the SD.

and secondary granules are produced during the maturation process and not in mature neutrophils, which contain granules carrying arginase-1 (Jacobsen et al, 2007). Markers of neutrophil maturity, for example, *MME* (CD10), *FCGR3A* (CD16), and *SELL* (CD62L), are all expressed less in immature CD45^low LDN in comparison with mature CD45^high LDN and NDN (Fig 7). We confirmed this difference in CD16 (Fig 4C) and CD10 (Fig 5D) expression at the protein level.

The transcriptomes of mature blood neutrophils (CD45^high LDN and NDN) are very similar, whether they are sourced from healthy donors or NSCLC patients (Figs 6 and 7, S5D, and S7). Mature CD45^high LDN and NDN express the highest levels of genes associated with chemotaxis, phagocytosis, and ROS biosynthesis (Fig 7).

In comparison with CD45^high LDN and NDN, TAN were slightly different in the PCA (Fig 6) and express lower levels of maturity marker CD10 and chemokine receptors CXCR1 and CXCR2 (Fig 7). Among genes that are more highly expressed in TAN, the vascular

endothelial growth factor A (*VEGFA*), which was reported to enhance the permeability of vascular endothelial cells and promotes tumor metastasis (Kim et al, 2017), stands out (Fig 7B).

## Discussion

In the blood of NSCLC and ovarian cancer patients, we identified two LDN populations that were distinguishable by the expression levels of CD45. The CD45^low LDN displayed immature morphology, and CD45^high LDN appeared mature. Only cells within the mature CD45^high LDN subset suppressed T-cell proliferation. Based on the expression of CD16 and CD10 (established markers for neutrophil maturity) (Marini et al, 2017), these two LDN subpopulations are analogous to the immature and mature neutrophil populations previously observed in the blood of head-and-neck cancer and melanoma patients (Lang et al, 2018; Gondois-Rey et al, 2021). We

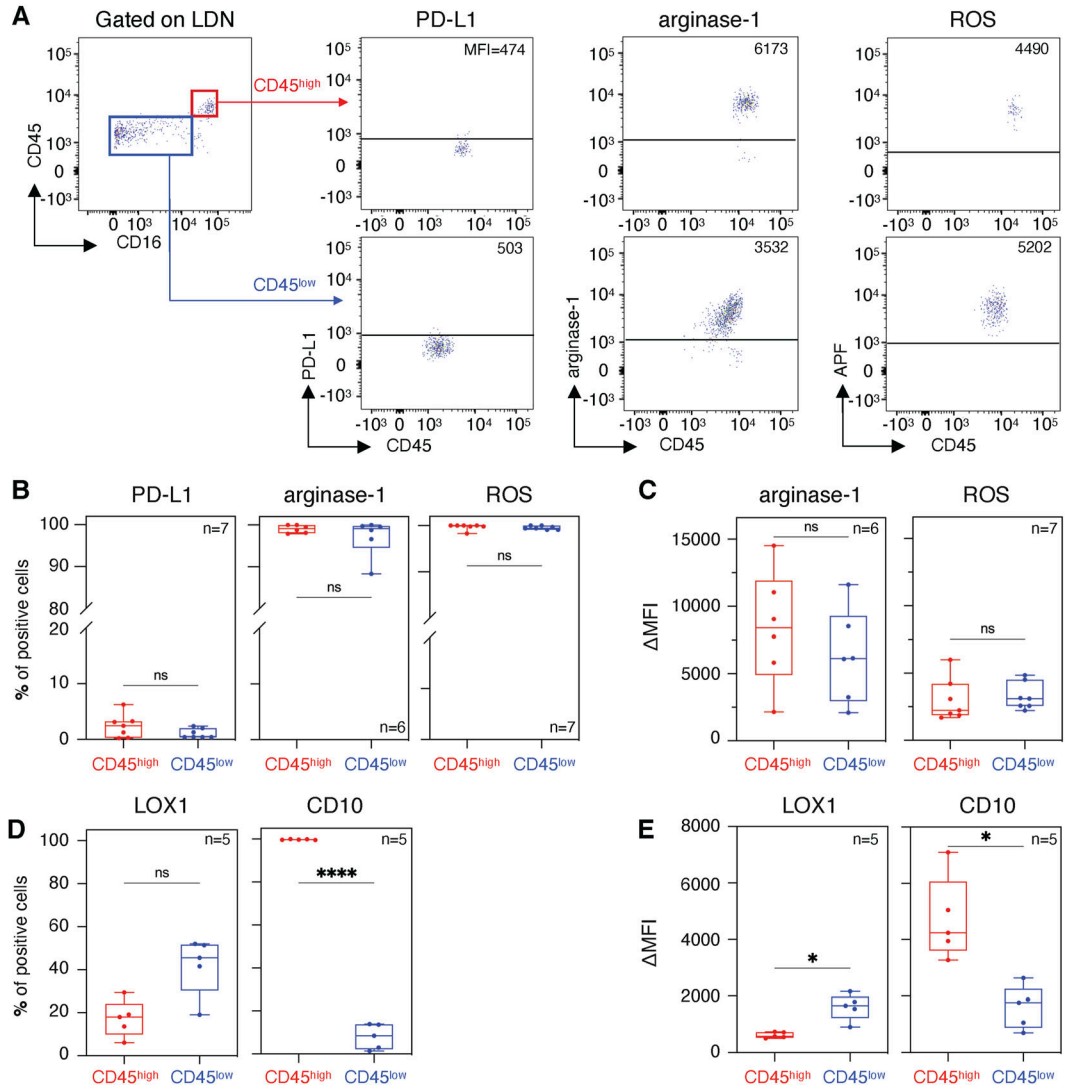

**Figure 5. Protein expression of PD-L1, arginase-1, LOX-1, and CD10 and detection of reactive oxygen species in CD45[high] low-density neutrophils and CD45[low] low-density neutrophils.**
**(A)** PD-L1 extracellular staining, arginase-1 intracellular staining, and reactive oxygen species detection by an aminophenyl fluorescein fluorescent probe. The horizontal line corresponds to the positivity threshold determined for each stain by its respective Fluorescence Minus One. **(B, C, D, E)** Percentage of cells above the positivity threshold and (C, E) the increase in mean fluorescence intensity between the stained sample and Fluorescence Minus One. Patient samples used in panels (B, C, D, E) were different. Boxes show the median, and whiskers, the minimal and maximal values. Statistical analyses: paired $t$ test. $P$-values: ns (non-significant), ≥0.05; *, <0.05; and ****, <0.0001.

conclude that the suppressive neutrophils in the blood of NSCLC and ovarian cancer patients are found in the population of mature neutrophils. When LDN in cancer patients exhibit suppressive activity, the NDN subset also displays suppressive properties. Only future experiments, such as single-cell transcriptomic analyses of the CD45[high] LDN and NDN, might reveal the homogeneity or heterogeneity of the suppressive subsets in detail.

Mature CD45[high] LDN closely resembled mature NDN both transcriptomically and functionally. In PCAs, mature CD45[high] LDN and NDN clustered together, with only 29 differentially expressed genes between patient-matched samples. In the clinic, elevated neutrophil-to-lymphocyte ratios in the blood of NSCLC patients are used to predict overall survival outcomes (Templeton et al, 2014;

Tashima et al, 2020). However, we observed no correlation between the neutrophil-to-lymphocyte ratios and the percentage of CD45[high] LDN in the blood (Fig S8). This leads us to conclude that elevated neutrophil numbers in the blood do not systematically lead to increased numbers of suppressive mature CD45[high] LDN.

Lectin-type oxidized LDL receptor-1 (LOX-1), immune checkpoint inhibitor PD-L1, arginase-1, and NOX2 have been proposed as markers of immunosuppressive LDN or mediators of LDN immunosuppression (Corzo et al, 2009; Rodriguez et al, 2009; Condamine et al, 2016; Gabrilovich, 2021; Tang et al, 2022). Surprisingly, we found that the genes encoding LOX-1, arginase-1, and NOX2 were less expressed in immunosuppressive mature NDN and CD45[high] CD16[high] LDN than in non-immunosuppressive immature CD45[low] CD16[low] LDN.

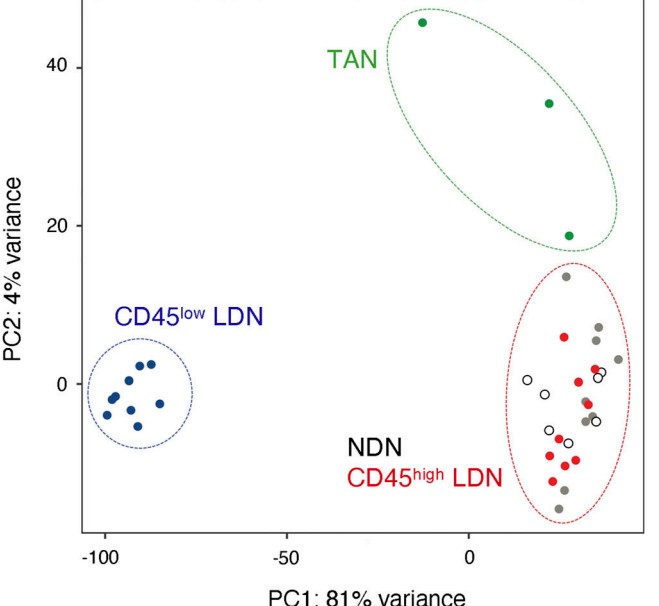

**Figure 6.  Tumor-associated neutrophils differ transcriptomically from CD45[high] low-density neutrophils and normal-density neutrophils from NSCLC patients.**
Principal component analysis of differentially expressed genes in neutrophil subtypes in NSCLC and healthy donors. Normalization of data and principal component analysis were performed with DESeq2 (R). Each point represents one sample (bulk population).

Protein levels appear to contradict the transcriptomic data, as intracellular staining for arginase-1 and ROS revealed similarly high levels in both CD45[high] and CD45[low] LDN. This could be explained by the maturation process of neutrophils: arginase-1 and NOX2 are stored in primary and secondary granules (Karlsson & Dahlgren, 2002). These granules are produced in earlier stages of neutrophil maturation. Therefore, it is reasonable to think that when the granules in maturing neutrophils have accumulated a sufficient amount of the respective proteins, the transcription of the corresponding genes is ceased (Jacobsen et al, 2007). Supporting this notion, a recent study comparing proteomic and transcriptomic data from NDN isolated from healthy donors found that protein presence and RNA expression were not always correlated (Hoogendijk et al, 2019).

An increased surface expression of PD-L1 was reported in mature and immunosuppressive LDN (Gondois-Rey et al, 2021). PD-L1 was also shown to mediate suppression of IFN-γ–stimulated neutrophils (de Kleijn et al, 2013). Here, we observed an increased expression of gene *CD274* (*PD-L1*) in mature immunosuppressive CD45[high] CD16[high] LDN, but PD-L1 surface expression was extremely low for both LDN subpopulations.

In comparison with mature CD45[high] LDN in blood, TAN clustered in a distinct population apart from blood NDN in PCAs of the transcriptomic data. We acknowledge that isolating the TAN involved enzymatic tissue digestion, which may have influenced the transcriptome of TAN. These TAN fell in between the metabolically more active immature CD45[low] blood LDN and the metabolically less active mature CD45[high] blood LDN and NDN. Particularly, genes associated with lipid metabolism and glycolysis were up-regulated, pointing to an adaptation to intratumoral hypoxia. We also observed that TAN up-regulated *VEGFA*, which enhances the permeability of vascular endothelial cells. This could be a response to a hypoxic environment and has been associated with a protumoral N2-TAN phenotype (Fridlender et al, 2009). *VEGFA* has been reported to promote tumor metastasis, and TAN were found to counteract anti-VEGF therapy in metastatic colorectal cancer (Stockmann et al, 2008; Kim et al, 2017; Schiffmann et al, 2019).

In conclusion, we observed in ovarian and lung cancer patients that only mature neutrophils exhibited suppressive properties, irrespective of their density. Interestingly, mature neutrophils with normal density isolated from non-cancerous blood donors did not display suppressive functions. The literature data are contradictory regarding the suppressive abilities of NDN: NDN from healthy individuals are sometimes described as suppressive (e.g., Westerlund et al, 2022). In other studies (e.g., Aarts et al, 2019), both NDN from healthy subjects and those from patients with head-and-neck carcinoma or mammary carcinoma inhibit T-cell proliferation only if they have been activated by the presence of formyl peptide fMLF. These discrepancies may stem from variations in the protocols used for isolating neutrophils, known for their fragility and sensitivity. In addition, differences might arise from varying priming or activation states depending on the patient and his pathology (Pillay et al, 2013). This could also imply that the suppressive functions of blood neutrophils are influenced by cues associated with the tumor microenvironment.

## Materials and Methods

### Human samples

Blood from non-cancerous donors (hemochromatosis patients) was collected at the Cliniques universitaires Saint-Luc (CUSL), Brussels, Belgium. Blood and solid tumor samples were collected before systemic treatment from ovarian cancer or non–small-cell lung cancer (NSCLC) patients during surgery. The samples were obtained from the CUSL with the approval of the ethical committee of CUSL (2017/11OCT/478—Belgian no: B403201734113; and 2019/17JUI/261—Belgian no: B403201941763), and this study was carried out in accordance with the principles expressed in the Declaration

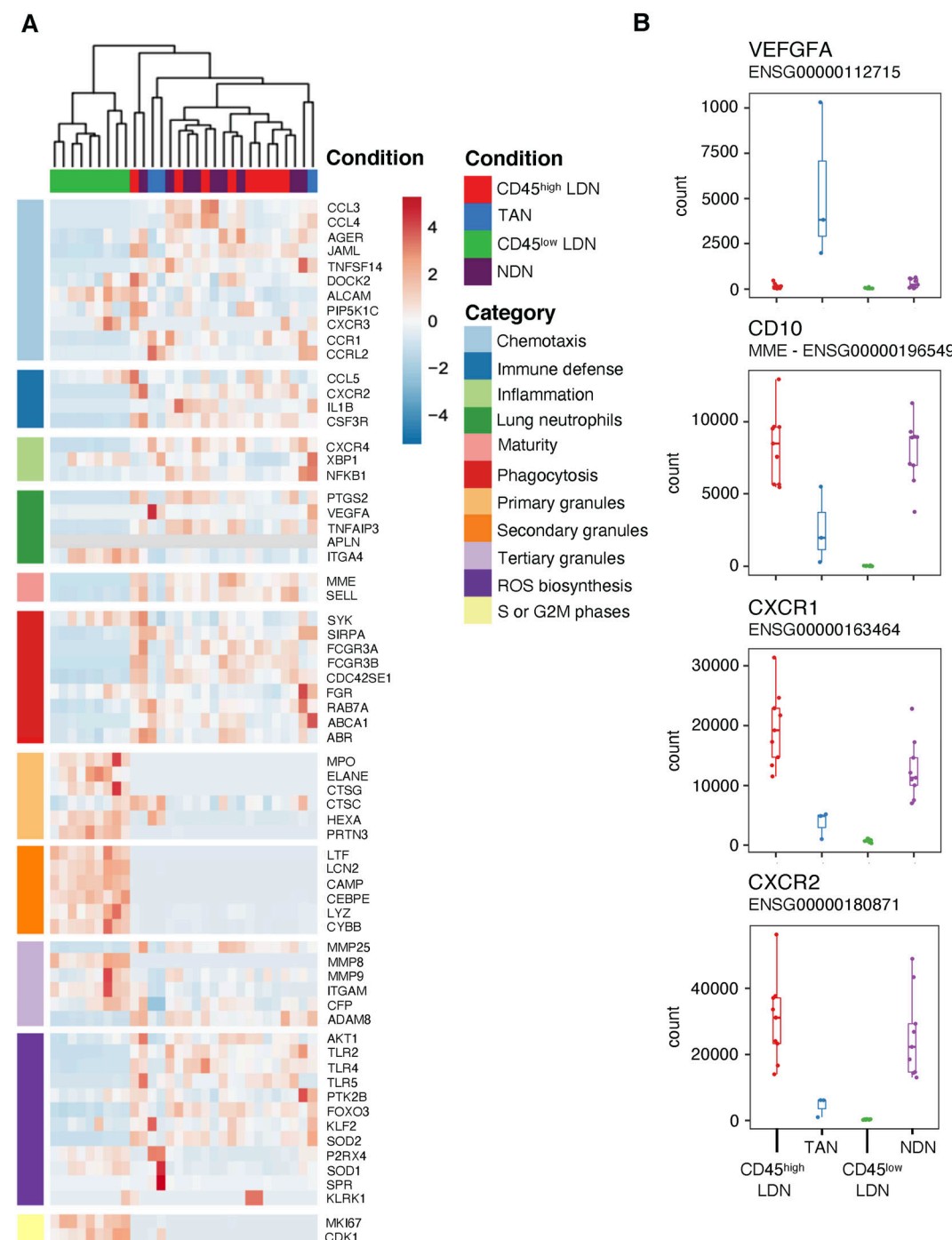

**Figure 7. Tumor-associated neutrophils differ in their expression of genes involved in chemotaxis and maturity compared with CD45^high low-density neutrophils (LDN) and normal-density neutrophils (NDN) from NSCLC patients.**
**(A)** Heatmap of selected genes that are differentially expressed in LDN and NDN subpopulations in NSCLC patients. **(B)** Comparison of read counts from the transcriptomic data for vascular endothelial growth factor A, CD10, CXCR1, and CXCR2 in LDN and NDN subpopulations in NSCLC patients. Boxes show the median, and whiskers, the 1,5x interquartile range.

of Helsinki. Patients gave their written informed consent, and their records were de-identified before the analysis. All samples were collected in the morning. The age of non-cancerous donors, ovarian cancer patients, and NSCLC patients was 59 ± 11, 61 ± 12, and 63 ± 11, respectively.

### Processing of tissue samples

Blood was diluted with PBS/1 mM EDTA (at RT) at a ratio of 4:1 to 1:2 before mononuclear cells were separated from erythrocytes and neutrophils by density gradient separation on Lymphoprep

(Fresenius/Axis-Shield). The gradient was established by centrifugation at 900$g$ for 20 min without acceleration or deceleration. The layer containing mononuclear cells is collected at the interface between the plasma and the Lymphoprep. These cells were washed three times with PBS/1 mM EDTA at 360$g$ for 10 min. The remaining platelets were removed by centrifugation at 330$g$ for 6 min. Neutrophils were recovered from the density gradient layer above the erythrocytes and further purified by density gradient separation with 1% m/v polyvinyl alcohol (Sigma-Aldrich) in 0,9% NaCl (Sigma-Aldrich). The remaining erythrocytes were lysed using 0.2% NaCl for 1 min. Physiological salt concentrations were restored with 1.2% NaCl. Solid tumors were mechanically dissociated with a MACS dissociator (Miltenyi) and enzymatically digested for 45 min at 37°C with 5 U/ml of DNase-I (DCLS; Worthington), 0.5 U/ml Liberase TL, and 0.26 U/ml Liberase DL (thermolysin/dispase-low, both from Sigma and Roche, catalogue #05401020001). The tumor suspension was washed with PBS/1 mM EDTA and passed sequentially through 70- and 40-$\mu$m filters. The light density fraction (Fig 1A) was isolated from the tumor cell suspension by density gradient separation using Lymphoprep as described above.

## Staining and antibodies for flow cytometry

Samples for flow cytometry were first incubated with Fc receptor blocking reagent (BD Biosciences) for 5 min at RT. Samples were then stained with antibodies and/or reagents (see below) for 15–20 min in the dark at RT to avoid temperature fluctuations and excessive centrifugation as much as possible to preserve neutrophil integrity. Stainings were performed in the presence of 1 mM EDTA and 2% human serum (HS, in-house). The samples were washed three times by centrifugation at 380$g$ for 7 min with PBS/1 mM EDTA supplemented with 2% HS, and resuspended in PBS/1 mM EDTA/2% HS for analysis. HS was prepared from pools of plasma from unselected ABO donors. Plasma was supplemented with $CaCl_2$ and bovine thrombin (ICN Pharmaceuticals) and allowed to clot for 3 h at 37°C. After centrifugation, serum was decomplemented (30 min at 56°C), filtered through 0.22-mm pore-size filters, and inactivated by gamma irradiation. Analyses were recorded using a FACS Fortessa flow cytometer (BD Biosciences), and sorting was performed on a FACSAria III cell sorter (BD Biosciences). All samples were analyzed with FlowJo (RRID: SCR_008520), v10. ROS presence was detected using aminophenyl fluorescein (Thermo Fisher Scientific). Cells were incubated for 30 min at RT with aminophenyl fluorescein at 50 $\mu$M, washed, and analyzed immediately on a FACS Fortessa flow cytometer (BD Biosciences) as described above. For arginase-1 intracellular staining, the cells were first stained with antibodies against surface molecules, fixed with BD Cytofix/Cytoperm (RRID: AB_2869008; BD Biosciences) for 20 min at 4°C, and washed twice by centrifugation at 380$g$ for 6 min at 4°C with BD Perm/Wash. The permeabilized cells were stained with an anti-arginase-1 antibody coupled to PE (clone A1exF5, RRID:AB_2734839; Thermo Fisher Scientific), washed twice by centrifugation at 380$g$ for 6 min at 4°C with BD Perm/Wash, resuspended in cold PBS/1 mM EDTA with 2% HS, and kept on ice before analysis on a FACS Fortessa flow cytometer. Antibodies and colorimetric stainings

used were as follows: eFluor 780 (1 in 1,000; Affymetrix), anti-CD19-BV510 (clone SJ25C1, RRID:AB_2737914; BD Biosciences), anti-CD56-BV510 (clone NCAM1, RRID:AB_2732786; BD Biosciences), anti-CD3-BV510 (clone HIT3a; BD Biosciences), anti-CD3-Alexa Fluor 488 (clone HIT3a, RRID:AB_493690; BioLegend), anti-CD33-BV711 (clone WM53, RRID:AB_2738045; BD Biosciences), anti-CD45-Alexa Fluor 700 (clone HI30, RRID:AB_493760; BioLegend), anti-CD14-PerCP (clone MoP9; BD Biosciences), anti-CD15-PE (clone HI98, RRID: AB_395802; BD Biosciences), anti-CD15-APC (clone W6D3, RRID: AB_756013; BioLegend), anti-CD16-BV421 (clone 3G8, RRID:AB_2716865; BD Biosciences), anti-CD206-FITC (clone 19.2, RRID:AB_394065; BD Biosciences), anti-HLA-DR-PE-Cy7 (clone L243, RRID: AB_493589; BioLegend), anti-CD11b-APC (clone ICRF44, RRID:AB_398456; BD Biosciences), anti-CD11c-PE-Cy7 (clone B-ly6, RRID:AB_10611859; BD Biosciences), anti-CD1a-BV421 (clone HI149, RRID:AB_2744316; BD Biosciences), anti-CD83-APC (clone HB15e, RRID:AB_398488; BD Biosciences), anti-CD3-PE (clone UCHT1, RRID:AB_314061; BD Biosciences), anti-CD8 alpha-APC (clone RPA-T8, RRID:AB_10896290; BD Biosciences), anti-arginase-1-PE (clone A1exF5, RRID:AB_2734839; Thermo Fisher Scientific), anti-PD-L1-APC (clone MIH2, RRID:AB_2749927; BioLegend), and 5-chloromethylfluorescein diacetate (CMFDA, C2925; Thermo Fisher Scientific) tracker dye. Rainbow beads (BD Biosciences) were added in proliferation assays at 2,500 beads per sample as counting and normalization beads.

## Assessing neutrophil-suppressive functions through T-cell proliferation assay

### Isolation of allogeneic monocytes and T cells

Monocytes and T cells from non-cancerous donors were isolated from PBMC by magnetic bead separation using autoMACS (Miltenyi). First, PBMC were separated into CD2-positive and CD2-negative fractions after 2 min of incubation with sheep erythrocytes (E-400; Institut Virion/Serion GmbH) at RT by Lymphoprep density gradient separation as described above. Monocytes were isolated from the CD2-negative population by positive selection using CD14 beads (Miltenyi) and further differentiated into mature dendritic cells (DC). T cells were isolated from the CD2-positive fraction by negative selection using CD56 beads (Miltenyi) and were frozen at −80°C in 50% RPMI, 40% FCS, and 10% DMSO for later use in the proliferation assay.

### Maturation of dendritic cells

Mature dendritic cells were differentiated from blood monocytes as previously described (Bruger et al, 2020). Briefly, monocytes (isolated from mononuclear cells of non-cancerous patients as described above) were resuspended in RPMI-1640 complemented with 10% FCS (Sigma-Aldrich), 200 nM L-alanyl-L-glutamine dipeptide (35050-038, GlutaMAX; Gibco), 100 U/ml penicillin + 100 $\mu$g/ml streptomycin (P4333; Sigma-Aldrich), 100 ng/ml of clinical grade GM-CSF (NDC0024-5843-05, Leukine [Sargramostim]; Sanofi-Aventis), and 30 ng/ml (5 × 10$^5$ U/ml) IL-4 (in-house) (DC medium). Monocytes were seeded in a 12-well tissue culture plate at 10$^6$ cells/well, and incubated for 5 d at 37°C and 5% $CO_2$. Dendritic cells were matured at Day 5 with 50 $\mu$g/ml Poly-IC (P1530; Sigma-Aldrich) and 10 ng/ml TNF-$\alpha$ (300-01A; PeproTech) in DC medium for another

2 d at 37°C and 5% $CO_2$. Cells were harvested at Day 7 and checked by flow cytometry to confirm their phenotype (CD3⁻, CD19⁻, CD14⁻, CD11c⁺, CD1a⁺, CD83⁺) (see antibodies for flow cytometry). Mature DC were frozen at –80°C in 50% RPMI-1640, 40% FCS, and 10% DMSO for later use.

### T-cell proliferation assay

Allogeneic T cells and mature DC were thawed in RPMI-1640 medium (Life Technology) complemented with 5% HS, 200 nM GlutaMAX, 100 U/ml penicillin and 100 μg/ml streptomycin (Pen/Strep), and 5 U/ml human recombinant IL-2 (CLB-P-476-700-1447; Novartis) (T-cell proliferation medium) complemented with 5 U/ml of DNase-I (DCLS; Worthington) for minimum 4 h at 37°C and 5% $CO_2$. A 96-well flat-bottomed plate (7007; Corning Costar) was coated with 1 μg/ml anti-CD3 antibody (clone OKT3, RRID:AB_571927; BioLegend) for a minimum of 4 h at 37°C. Allogeneic and autologous T cells were labeled with 0.5 μM CMFDA in RPMI-1640 without serum for 20 min at 37°C and 5% $CO_2$, and then washed once with RPMI-1640 by centrifugation at 380$g$ for 8 min. Labeled (allogeneic or autologous) T cells were seeded at a final concentration of 0.6 × 10⁶ cells/ml in the T-cell proliferation medium. Mature DC were seeded at a final concentration of 2 × 10⁴ cells/ml in the T-cell proliferation medium into wells containing allogeneic T cells only. Co-stimulation was provided by anti-CD28 antibodies (purified NA/LE mouse-anti-human, RRID:AB_396068; BD Biosciences, final concentration 2 μg/ml in the T-cell proliferation medium). Sorted myeloid cells were resuspended in the T-cell proliferation medium and seeded at final concentrations of 0.2 × 10⁶, 0.066 × 10⁶, or 0.02 × 10⁶ cells/ml. The final volume per well was 200 μl. The cells were incubated for 4 d at 37°C and 5% $CO_2$. T-cell proliferation was assessed by tracker dye dilution using flow cytometry. When the T-proliferation assay was performed with autologous T cells, T cells, LDN, and neutrophils were isolated from the same blood sample and used immediately. The division index indicates the average number of divisions a T cell within the well has experienced.

### Cytospin analysis

After isolation by flow cytometry, LDN (CD45^low or CD45^high) from tumors and blood, and NDN from blood were centrifuged and resuspended at >50,000 cells/ml in a minimum volume of 100 μl of FCS. The cell suspension was processed within 6 h of collection using the cytospin technique. First, the cell concentration was evaluated using the XN-1000 body fluid analyzer (Sysmex). The average cell concentration was 50 cells/μl. Then, a cytospin smear was made using the cytospin technique. The slides, filtercards, and Cellfunnels were placed in the appropriate slots of the metal housing. For each sample, 100 μl was placed in the well and the cytospin construction was spun for 4 min at 28$g$ using the Cellspin I cytocentrifuge (THARMAC). The slides were quickly removed from the cytospin construction and air-dried. Finally, they were colored with the May–Grünwald–Giemsa stain using the Aerospray Hematology Pro slide stainer (ELITechGroup). The slides were analyzed visually (objective lens x100).

### RNA-sequencing and transcriptome analyses

RNA-seq libraries were constructed using Nextera XT DNA Library Preparation Kit (FC-131-1096; Illumina) in combination with Nextera XT Index Kit v2 Set D (FC-131-2004; Illumina). The construction was unstranded and poly-A–based. RNA libraries were sent for ultra-low input RNA sequencing to Genewiz (Leipzig). Between 20 and 30 million paired reads were sequenced per sample on an Illumina HiSeq 2 × 150 platform. The quality of the reads was controlled by FastQC (Babraham Bioinformatics) before and after trimming with Trimmomatic (RRID:SCR_011848) (0.38 version [Bolger et al, 2014]). The reads were aligned to the human genome GRCh38 using the HISAT2 program (Kim et al, 2019). SAMtools (RRID:SCR_002105) was used to index the reads in .bam files. Aligned reads were counted with the featureCounts program from the Subread package (Liao et al, 2014). Normalization, PCA, and differential expression analysis were realized on R using the DESeq2 package (RRID:SCR_000154) (Love et al, 2014). Genes were pre-filtered to include all genes with >10 reads in at least three different samples. For differentially expressed genes, a log₂ fold change threshold was set to >2 (min. fourfold) with a $P$-value below 0.05. Volcano plots were drawn using the Enhanced Volcano Plot package (R). Potential differentially expressed cell-surface proteins were selected based on the following gene ontology annotations: GO:0005886, GO:0004872, and GO:0009897.

### Statistical analyses

The frequencies of granulocytic cells among the total myeloid population (CD33⁺) or hematopoietic cells (CD45⁺) were determined using FlowJo. The normality of distribution was assessed by the Shapiro–Wilk test. For proliferation assays, the division index, the number of cell cycles, and the number of cells that undergo division at Day 0 were determined using the mean fluorescence intensity of CMFDA as described by FlowJo. All statistical tests were made on R or GraphPad Prism (version 10.0.2) and were indicated in the figure legends.

## Data Availability

Bulk RNA-seq raw data were deposited into the Gene Expression Omnibus database under accession number GSE234172. All other relevant data supporting the key findings of this study are available within the article and its Supplementary Information files or from the corresponding author upon reasonable request.

## Supplementary Information

# Acknowledgements

We thank the cancer patients for taking part in this study. We also thank the surgeons Valerie Lacroix and Delphine Hoton (Cliniques universitaires Saint-Luc, Brussels, Belgium). We appreciate the work done by Nick van Gastel for his metabolic analysis of our transcriptomic data. We thank Isabelle Grisse for editorial assistance. This work was supported by WALinnov grant from the Walloon region, Belgium (program IMMUCAN 1610119), the Fonds de la Recherche Scientifique (FNRS), Belgium, Credit de recherche FNRS (grant number: J.0209.20), Programme de recherche FNRS (PDR T.0226.23), and Belgian Foundation against Cancer (grant number: 2022-186). C Vanhaver and N Delhez were supported by the Walloon Region, Belgium (FRIA/FNRS). AM Bruger was supported by a TELEVIE fellowship and a de DUVE fellowship. F Aboubakar Nana was supported half-time by the FNRS (Belgium) as "specialiste post-doctorant" (convention 1R90123F). We thank the de DUVE Institute (Brussels) and European COST Action 20117 Mye-InfoBank for their continuous support.

## Author Contributions

C Vanhaver: conceptualization, data curation, formal analysis, investigation, visualization, methodology, and writing—original draft.
F Aboubakar Nana: conceptualization, resources, data curation, supervision, and writing—review and editing.
N Delhez: formal analysis, methodology, and writing—original draft.
M Luyckx: resources.
T Hirsch: supervision, investigation, methodology, and writing—original draft.
A Bayard: formal analysis, investigation, and methodology.
C Houbion: formal analysis, investigation, and methodology.
N Dauguet: formal analysis and investigation.
A Brochier: resources.
P van der Bruggen: conceptualization, supervision, funding acquisition, project administration, and writing—original draft, review, and editing.
AM Bruger: data curation, formal analysis, supervision, investigation, methodology, and writing—original draft, review, and editing.

## Conflict of Interest Statement

The authors declare that they have no conflict of interest.

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
