## [Reviewer comments · Life Science Alliance]

Life Science Alliance

Immunosuppressive Low-Density Neutrophils in the Blood of Cancer Patients Display a Mature Phenotype

Christophe Vanhaver, Frank Aboubakar Nana, Nicolas Delhez, Mathieu Luyckx, Thibault Hirsch, Alexandre Bayard, Camille Houbion, Nicolas Dauguet, Alice Brochier, Pierre van der Bruggen, and Annika Bruger

DOI: <https://doi.org/10.26508/lsa.202302332>

Corresponding author(s): *Pierre van der Bruggen, de Duve Institute*

Review Timeline:

Submission Date:	2023-08-23
Editorial Decision:	2023-10-02
Revision Received:	2023-10-09
Editorial Decision:	2023-10-11
Revision Received:	2023-10-26
Accepted:	2023-10-27

Scientific Editor: *Eric Sawey, PhD*

Transaction Report:

October 2, 2023

Re: Life Science Alliance manuscript #LSA-2023-02332-T

Prof. Pierre van der Bruggen
Ludwig Institute for cancer research
de DUVE Institute
74 avenue Hippocrate
UCL B1.74.03
Brussels 1200
Belgium

Dear Dr. van der Bruggen,

Thank you for submitting your manuscript entitled "Immunosuppressive Low-Density Neutrophils in the Blood of Cancer Patients Display a Mature Phenotype" to Life Science Alliance. The manuscript was assessed by an expert reviewer, whose comments are appended to this letter. We invite you to submit a revised manuscript addressing the Reviewer comments.

When submitting the revision, please include a letter addressing the reviewer comments point by point.

Thank you for this interesting contribution to Life Science Alliance. We are looking forward to receiving your revised manuscript.

Sincerely,

B. MANUSCRIPT ORGANIZATION AND FORMATTING:

Reviewer #1 (Comments to the Authors (Required)):

In their research manuscript, Vanhaver et al isolated neutrophils from blood or tumor of patients with NSCLC or ovarian cancer, as well as from blood of non-cancer patients, and investigated their propensity to act as T-cell immunosuppressive neutrophils in vitro. Importantly, they separated NDNs from LDNs for these functional tests. Their data elegantly concluded that not only LDN (agreeing with the literature) but also NDN from patients displayed immunosuppressive activity. This is an important finding, because it challenges the current view that only LDN suppress T cell proliferation.

Further, they identified two subsets of LDNs based on differential expression of CD45. They analysed differences in gene and protein expression between these two subsets, concluding that select genes were differentially expressed, which was not reflected at the protein level. This is important, as many studies use gene expression comparisons for innate immune cell subsets, not necessarily proteins.

They then compared transcriptomes of NDN, LDN(CD45high), LDN(CD45low) and TANs, uncovering a high degree of similarity and only few molecular changes, when considering bulk samples.

This is a good investigation that advances our knowledge about neutrophil phenotypic and functional diversity in cancer. The data are strongly supportive of each main point of the paper and I only have minor comments.

Minor comments:

1. For the T-cell immunosuppression part, the authors conclude that both LDN and NDN can be immunosuppressive. Reviewer notices (especially from the co-culture with autologous T cells) that NDN are invariably, and potently immunosuppressive, while LDN can be, but not always. Thus, it seems clear that NDN, at least in NSCLC, are more potent at Tcell immunosuppression in comparison to LDN. Although the authors cautiously state that "In fact, the suppressive activity of NDN was often more pronounced." (Page 5), could they more firmly conclude and discuss their findings in relation with current literature, in the Discussion?
2. Do the authors know if the protein expression of PDL1, ARG1, NOX2 and ORL1 is different comparing NDN to LDN? Maybe this information can be found in the literature, and would be nice to comment on, especially that there could be discrepancies between transcript and protein, as the authors highlighted when comparing LDN(CD45high) to LDN(CD45low).
3. Figure 1 legend: Remove "Data using autologous T-cells are shown in Sup." in "Data using autologous T-cells are shown in Sup. Data using autologous T-cells are shown in Sup. Figure 2 D."
4. Text p. 5. Reviewer feels that the following sentence should be split into two: "These frequencies are consistent with the frequencies reported (Cassetta et al., 2020) LDN yields from ovarian cancer blood samples were often too low (1,000-10,000 cells) for further functional experimentation."

Brussels, the 6th October 2023

Dear Editors,

We are submitting for your consideration a **revised** manuscript entitled “**Immunosuppressive Low-Density Neutrophils in the Blood of Cancer Patients Display a Mature Phenotype**” by Christophe Vanhaver et al.

The questions/comments of reviewers are written in black, our answers are written in blue. The modifications are indicated as track changes in the text.

Reviewer #1

Minor comments:

1. For the T-cell immunosuppression part, the authors conclude that both LDN and NDN can be immunosuppressive. Reviewer notices (especially from the co-culture with autologous T cells) that NDN are invariably, and potently immunosuppressive, while LDN can be, but not always. Thus, it seems clear that NDN, at least in NSCLC, are more potent at Tcell immunosuppression in comparison to LDN. Although the authors cautiously state that "In fact, the suppressive activity of NDN was often more pronounced." (Page 5), could they more firmly conclude and discuss their findings in relation with current literature, in the Discussion?

We have replaced “often” by “systematically” in order to more firmly conclude.

We added also a few sentences to the discussion to compare our data with the literature.

2. Do the authors know if the protein expression of PDL1, ARG1, NOX2 and ORL1 is different comparing NDN to LDN? Maybe this information can be found in the literature, and would be nice to comment on, especially that there could be discrepancies between transcript and protein, as the authors highlighted when comparing LDN(CD45high) to LDN(CD45low).

A supplementary figure (Sup. Figure 4) has been added containing the data requested. We have also described these new data in the text

3. Figure 1 legend: Remove "Data using autologous T-cells are shown in Sup." in "Data using autologous T-cells are shown in Sup. Data using autologous T-cells are shown in Sup. Figure 2 D."

The modification has been introduced.

4. Text p. 5. Reviewer feels that the following sentence should be split into two: "These frequencies are consistent with the frequencies reported (Cassetta et al., 2020) LDN yields from ovarian cancer blood samples were often too low (1,000-10,000 cells) for further functional experimentation."

The modification has been introduced.

Best regards,

Prof. Pierre van der Bruggen

October 11, 2023

RE: Life Science Alliance Manuscript #LSA-2023-02332-TR

Prof. Pierre van der Bruggen
de Duve Institute
TILS
avenue Hippocrate 75
Brussels 1200
Belgium

Dear Dr. van der Bruggen,

Thank you for submitting your revised manuscript entitled "Immunosuppressive Low-Density Neutrophils in the Blood of Cancer Patients Display a Mature Phenotype". We would be happy to publish your paper in Life Science Alliance pending final revisions necessary to meet our formatting guidelines.

- please upload all figure files as individual ones, including the supplementary figure files; all figure legends should only appear in the main manuscript file after the references section
- please add an ORCID ID for the corresponding author--you should have received instructions on how to do so
- please add the Twitter handle of your host institute/organization as well as your own or/and one of the authors in our system
- please upload the clean version of the manuscript file without track changes
- please use the [10 author names et al.] format in your references (i.e., limit the author names to the first 10)
- figure 6 has only one panel; there is no need to label it as A. Please correct
- there is a callout for Figure S4D, and this figure has only panels A and B; please correct
- please add callouts for Figures 5A; S2D; S4A,B; S5A-D and S8 to your main manuscript text
- please include the GEO accession number for the RNA-seq data in the Data Availability statement at this point

A. FINAL FILES:

B. MANUSCRIPT ORGANIZATION AND FORMATTING:

Sincerely,

October 27, 2023

RE: Life Science Alliance Manuscript #LSA-2023-02332-TRR

Prof. Pierre van der Bruggen
de Duve Institute
TILS
avenue Hippocrate 75
Brussels 1200
Belgium

Dear Dr. van der Bruggen,

Thank you for submitting your Research Article entitled "Immunosuppressive Low-Density Neutrophils in the Blood of Cancer Patients Display a Mature Phenotype". It is a pleasure to let you know that your manuscript is now accepted for publication in Life Science Alliance. Congratulations on this interesting work.

DISTRIBUTION OF MATERIALS:

Again, congratulations on a very nice paper. I hope you found the review process to be constructive and are pleased with how the manuscript was handled editorially. We look forward to future exciting submissions from your lab.

Sincerely,
